# Sonar Image Target Detection Based on Style Transfer Learning and Random Shape of Noise under Zero Shot Target

**Jier Xi** ⬥, **Xiufen Ye** *⬥ **and Chuanlong Li**

College of Intelligent Systems Science and Engineering, Harbin Engineering University, Harbin 150001, China
* Correspondence: yexiufen@hrbeu.edu.cn

**Abstract:** With the development of sonar technology, sonar images have been widely used to detect targets. However, there are many challenges for sonar images in terms of object detection. For example, the detectable targets in the sonar data are more sparse than those in optical images, the real underwater scanning experiment is complicated, and the sonar image styles produced by different types of sonar equipment due to their different characteristics are inconsistent, which makes it difficult to use them for sonar object detection and recognition algorithms. In order to solve these problems, we propose a novel sonar image object-detection method based on style learning and random noise with various shapes. Sonar style target sample images are generated through style transfer, which enhances insufficient sonar objects image. By introducing various noise shapes, which included points, lines, and rectangles, the problems of mud and sand obstruction and a mutilated target in the real environment are solved, and the single poses of the sonar image target is improved by fusing multiple poses of optical image target. In the meantime, a method of feature enhancement is proposed to solve the issue of missing key features when using style transfer on optical images directly. The experimental results show that our method achieves better precision.

**Keywords:** sonar image; style transfer; random shape of noise; multiple poses; feature enhancement



## 1. Introduction

With the improvement of sonar equipment, sonar images have gained a high level of achievement in underwater exploration [1] and target detection [2,3]. Compared to limitation of optical sensor in target detection, such as short detection distance, bad underwater visibility, and so on, side-scan based sonar target detection methods are widely used and are more effective in terms of distance and visibility.

Now and then, the real situation is much more complicated, as when images must be detected, and there is no training data available [4]. At present, the deep convolutional neural network (DCNN) is widely used in the sonar target detection [5,6]. Many scholars have been studying side-scan sonar-based object detection [7,8] with DCNN. In the meantime, sonar image detection based on noise analysis has also developed [9,10]. It has highly improved sonar detection accuracy compared with traditional recognition methods.

Nevertheless, the high cost of underwater experiments [11], such as target deployment underwater, the diversity of sonar devices, and the search for suitable experiment area, etc., has caused a lack of samples. Applicable results of sonar image detection have received much attention. However, given the minimal training data, few systems are widely used in real applications [12,13].The characteristics of a complex underwater environment and the lack of samples limit the generalization ability and precision of the sonar object detection.

Therefore, this paper comprehensively considers the complex underwater situation and lack of samples. To begin with, we use style transfer [14] optical images to augment pseudo samples [4]. However, the limitation of a small area on images with style transfer will cause low performance on object detection. To improve the performance of sonar target detection, we propose shape noises on images to solve the situation of mud and

sand obstruction and a mutilated target in the real environment. Furthermore, we combine various optical image datasets to enhance multiple poses to solve the single-state issue of targets. Finally, considering the key features of reflector and shadow are missing when we use style transfer directly, we use the binary and gamma methods [15,16] to enhance object features via frequency analysis on real sonar images. The remainder of this paper is organized as follows.

Section 2 introduces the related work about existing method and shortages.

Section 3 introduces our methods, including the data augmentation and simulation methods based on feature enhancement and the addition of random shape noises.

Section 4 introduces the comparison of the existing methods and their training results with our experiments and compares the results of our designed experiments.

## 2. Related Works

To overcome the shortage of samples, scholars have been working on zero-shot methods [2–4] to augment samples. The fine-tuning of a pretrained CNN is a useful method in sonar image detection [2,3]. Lee et al. [4] adopted StyleBankNet [14] to perform style transfer simulations of optical images of the human body, further improving sonar object detection and attained an 86% precision. The samples are generated by the software of computer aided design (CAD), but still require large simulation work to generate samples. Li et al. [3] made full use of the style transfer whitening and coloring transform (WCT) method and the remote sensing image simulation sonar image for target style transfer. It is effectively applied to underwater sonar image object detection which obtained 87.5% of precision. This method applied a large number of remote sensing images to transfer sonar images as sample data for training a DCNN model. However, it cannot express features of a target properly without considering the image environment (the state of target, such as target damage and corrupt, target postures, etc.). Yu, Yongcan et al. [17], by using transformer-YOLOv5, attained an accuracy of 85.6%. Huang et al. [2] combined a 3D model, amplified data, equipment noise, and image mechanism to extract target features and simulate target damage and postures via DCNN and a fine-tuning style transfer method. The method achieved 85.3% precision and 94.5% recall. Song et al. [18] proposed an efficient sonar segmentation method based on speckle noise analysis, which facilitated pixel-wise classification, and a single-stream deep neural network (DNN) with multiple side-outputs to optimize edge segmentation.

Most of the studies focus on amplifying samples and image mechanism, without considering too much on the target's real environment, like mud and sand obstruction, miss target parts, multiple state of the target, and shadow and reflector on real sonar data. In order to solve the problem, this paper proposes a method based on a fast-style learning [19] and random shape noise model, and simultaneously integrates multiple morphological fusions of optical targets to improve the uncertainty of target detection. In addition, for getting closer to real data and enhancing the object features, an image-processing [15,16] method based on binary and gamma transformation simulated shadow and reflector of object in training data. By adding random shape noise to the target image, the optical image is simulated for sediment cover and mutilated targets. The multiple state of optical image targets is integrated to deal with the problem of multiple forms of underwater targets, and the detection rate of sonar images is further strengthened.

## 3. Our Method

### 3.1. Problems Definition and Our Framework

As described in Section 2, a lack of samples is a common problem in detecting sonar images, which leads to low model performance. Most of the existing methods come from transfer optical data to rising sonar detection performance [2–4] with less consideration of the real underwater environment. Combined the issue, the crux of deep learning work is to prepare datasets. To extract complex features from the dataset and based on zero-shot samples in target detection, in this section we present the main contributions of the paper.

We consider three major aspects in dataset design. (1) We define a dataset and augmentation from an optical image to extend multiple poses on the target. (2) We transfer the optical image to sonar style. (3) We design random shape noise on a target to simulate mud and sand obstruction.

In our experiment, the data processing to generate training data is shown in Figure 1.

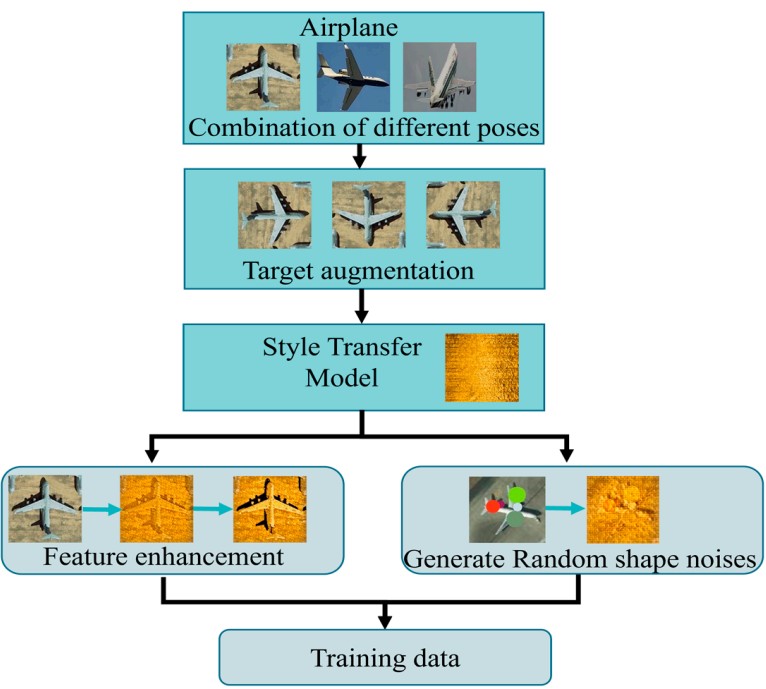

**Figure 1.** Data processing to generate training data.

The overall process framework of our main method is shown in Figure 2. It includes three parts, data preprocessing, style transfer, and detection model training. Data preprocessing integrates different optical image datasets into target categories, such as airplanes, boats, cars, etc., where sediment occlusion on the seabed is simulated by adding random noise.

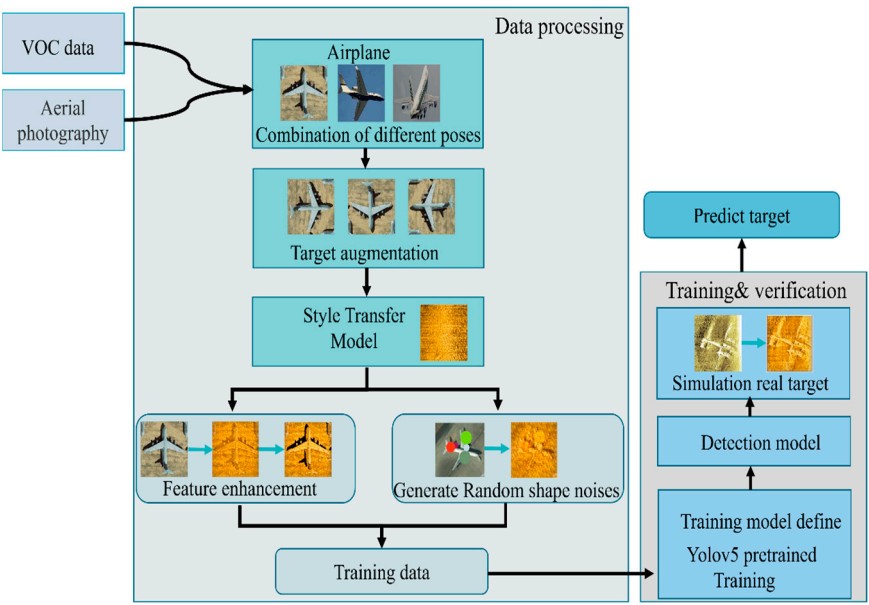

**Figure 2.** Our process framework.

A stylized model with sonar background images to stylize preprocessed and integrated datasets. Different sonar datasets have different image textures, and style transfer learning method is pays attention to image texture. To reduce the manual work in different sonar datasets, the proposed method can also be applied to different sonar datasets. The stylized dataset is then rotated and stretched by data preprocessing to further enhance the target final attitude dataset. We enhance the key features via primary simulation.

The processed image is trained in combination with the yolov5 model to obtain an image model with detection sonar. The real sonar data is styled before the scene is restored and detected.

### 3.2. Improved Methods of Data Combination and Augmentation

Most of the existing methods focus on amplifying samples with fewer combined target poses. The final state (upright, inverted, incline, etc.) of real targets are always uncertain. The example of real sonar images with different poses are shown in Figure 3.

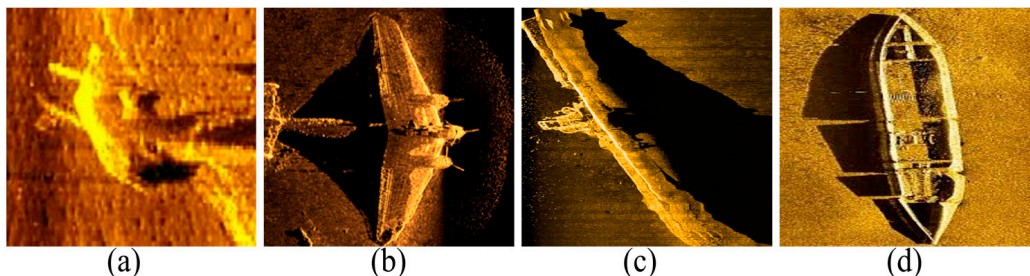

(a)       (b)       (c)       (d)

**Figure 3.** Real sonar images (**a**) Airplane with incline pose. (**b**) Airplane right pose. (**c**) Shipwreck with lateral pose. (**d**) Shipwreck with frontage pose.

Because the uncertain object state to be detected on the seabed will eventually form, we propose a method that uses a large amount of optical data with different poses to help enrich the final form of the object. For example, with regard to aircraft data, there are different kinds of postures as interpreted by our senses, but cognitively they are all considered to be airplanes, as shown in Figure 4.

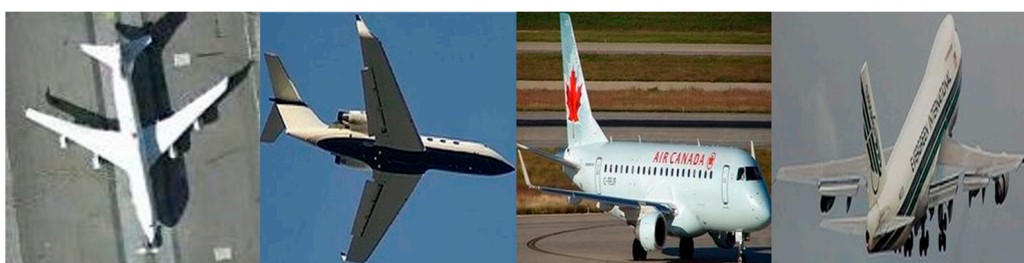

**Figure 4.** Multiple attitudes of the optical image on same target.

To improve multiple target morphologies, we have approached the data augmentation method [20] in this paper. The extraction of image features in terms of stretching, scaling, and rotation for sonar object detection. The image size of targets in training data were set to (64, 64), (64, 32), (32, 32), (32, 64), and (128, 128). The target size can be adjusted to be larger, which can be matched with the background image. The generated random position and the target merge to the background of sonar image. The example of image expansion is shown in Figure 5.

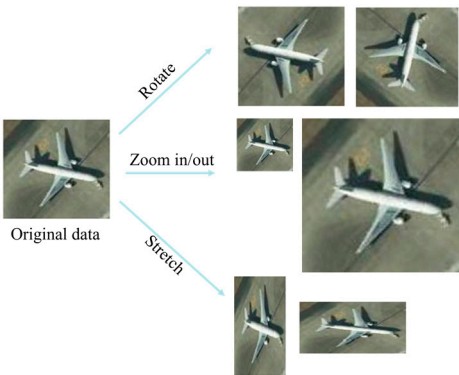

**Figure 5.** Data expansion in one target.

The amount of original and augmentation data is shown in Table 1 where org-set is the number of images integrated from the original different optical data sets, and augmentation dataset is the number of enhanced data.

**Table 1.** Amount of original and augmentation data by target type.

| Segment No. | Class | Org Dataset | Augmentation Dataset |
| --- | --- | --- | --- |
| 1 | Aeroplane | 739 | 1591 |
| 2 | Bicyle | 254 | 557 |
| 3 | Car | 781 | 1698 |
| 4 | Person | 650 | 1390 |
| 5 | Ship | 583 | 1289 |

### 3.3. Improved Methods of Style Transfer on Image Dataset

The zero shot samples are a universal issue in underwater target detection. Many scholars have been studying on transfer learning and data enhancement, which is descried in Section 2. The performance of style transfer-based models in sonar target detection has improved significantly. It comes to be a technical trend of sonar target detection. The original image and transfer result is shown in Figure 6.

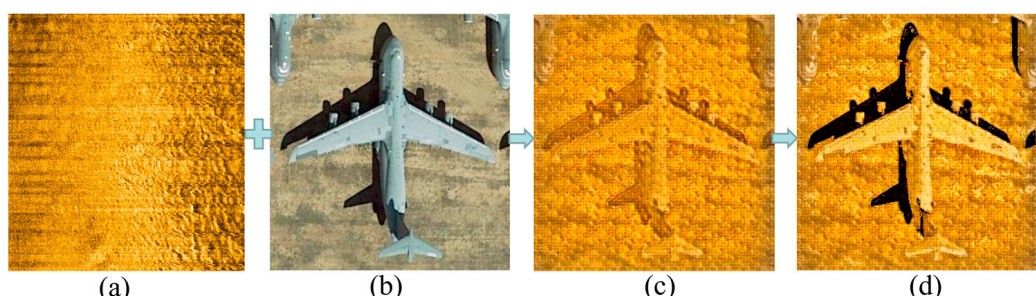

     (a)            (b)            (c)            (d)

**Figure 6.** Original sonar image with optical target transfer to sonar style target. (**a**) Original sonar image. (**b**) Optical image. (**c**) Traditional style transfer image. (**d**) Feature enhancement style transfer image.

Generally, style transfer can be described as being in two steps. First, the style transfer network via style image and content images generate a style model. Secondly, we input an image into the generated model and output the styled image.

By using the style transfer method directly, the key feature of shadow and reflector in sonar images will be lost, and most scholars have little regard for the features when using style transfer. A reference example is shown in Figure 7.

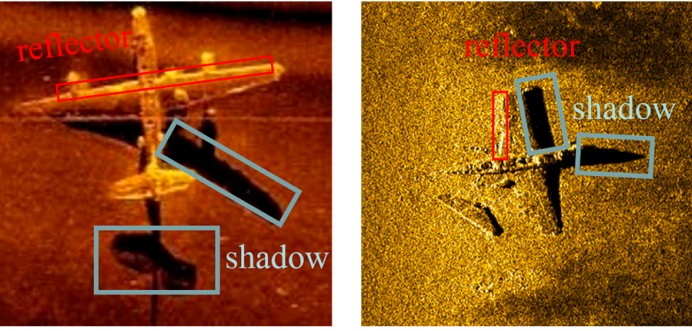

**Figure 7.** Example of shadow and reflector on real sonar target.

The network of the object-detection model extracts high-frequency signals as "key" features in the sonar image. Mostly, Fourier transform (FT) and inverse Fourier transform (IFT) are the main analysis methods in image frequency. An example of original sonar images of feature distribution on the frequency area is shown in Figure 8b, and the Gaussian filter of the frequency feature is shown in Figure 8c.

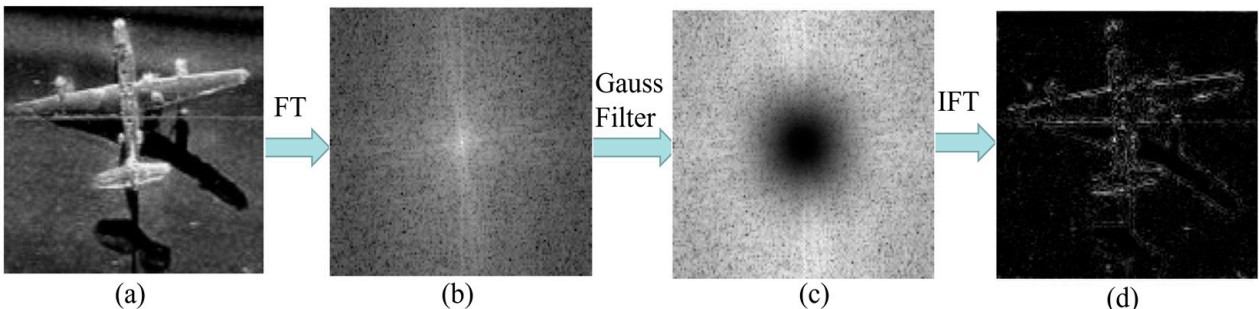

**Figure 8.** Frequency analysis on original sonar image. (**a**) Gray image on original sonar target. (**b**) Fourier transform on sonar image. (**c**) Gaussian filter frequency in frequency image. (**d**) Gaussian filtered image.

An example that uses the style transfer method on an optical target directly is shown in Figure 9.

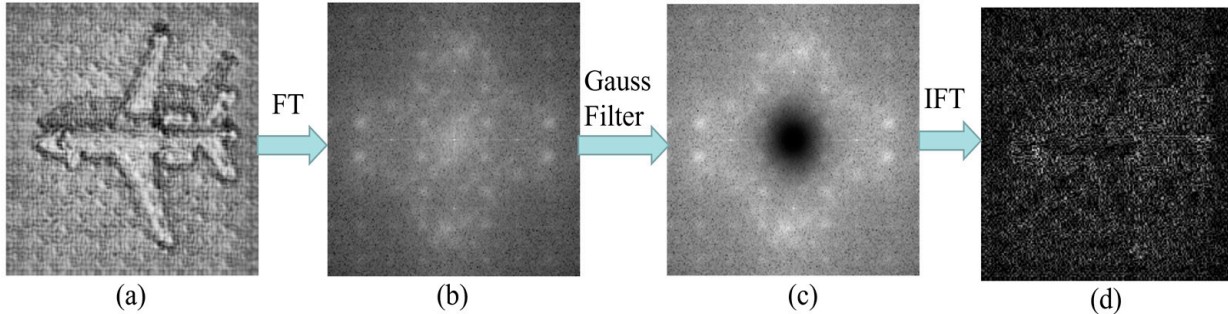

**Figure 9.** Frequency analysis on style transfer sample image. (**a**) Gray image using style transfer directly. (**b**) Fourier transform on sonar image. (**c**) Gaussian filter frequency in frequency image. (**d**) Gaussian filtered image.

Apparently, the distribution of a high-frequency signal is not smooth, according to Figure 9b, and the object feature are unclear according to Figure 9d.

To enhance the features, we proposal a simulation method that can enhance the features based on fast style transfer [19] as shown in Figure 10, and an analysis example of our method is shown in Figure 11.

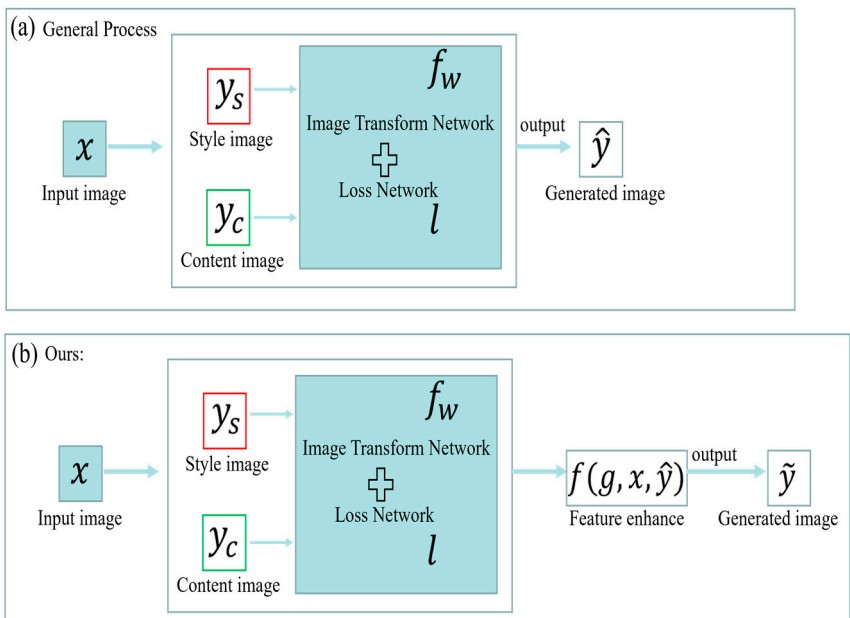

**Figure 10.** Style transfer process. (**a**) General style transfer process. (**b**) Enhancement feature based on style transfer.

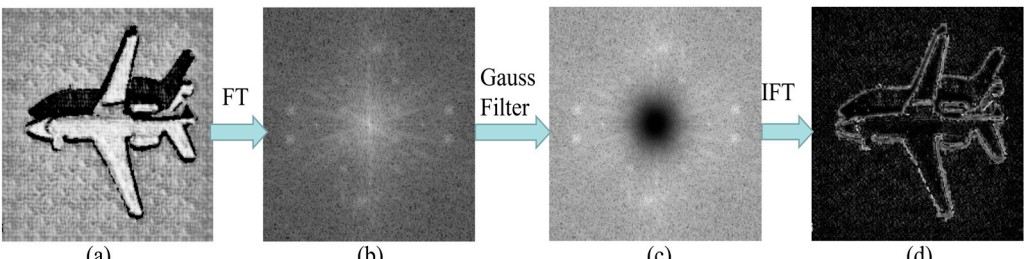

**Figure 11.** Frequency analysis on enhancement style transfer sample image. (**a**) Gray image on using style transfer directly. (**b**) Fourier transform on sonar image. (**c**) Gaussian filter frequency in frequency image. (**d**) Gaussian filtered image.

The general process of style transfer is shown in Figure 10a. Here, $x$ is the content image, $f_w$ is the network of style transfer, $y_c$ is $x$, and $y_s$ is the style image. Then, $\hat{y}$ is the generated image which is input image $x$ via style transferred by the network of $f_w$. The content of $\hat{y}$ is similar to $y_c$. The style of $\hat{y}$ is similar to $y_s$. The mathematical principles explanation is below. We define $p$ as sytle image, $a$ as content image, that is, to be style transferred. For example, $p$ is a backgournd of sonar image, $a$ is an optical image, $f$ is the transferred image that has sonar image style. We define two lost functions, $L_{style}$ and $L_{content}$. $L_{style}$ expects $f$ to be more similar to $p$ in terms of style. $L_{content}$ expects $f$ to be more similar to $a$ in terms of content, which is shown in formula [19]:

$$l(a, f, p) = \alpha * l_{style}(p, f) + \beta * l_{content}(a, f). \tag{1}$$

Based on the general process of style transfer, which is shown in Figure 10a, we define a function $f(g, x, \hat{y})$ to enhance the target's shadow and reflector in Figure 10b. Here, $x$ is original image, $\hat{y}$ is generated result from Figure 10a, $g$ is enhancement function which is implemented by binary and gamma transformation processing [15,16]. It can be expressed as follows:

$$g(B(x, \theta_1, \theta_2), \gamma), \theta_1 < \theta_2 \in [0, 255], \gamma \in [0, 15]. \tag{2}$$

Here, $B$ is a binarization function, and $\theta_1, \theta_2$ is the threshold value, $\gamma$ is the threshold value of gamma function, and $\widetilde{y}$ is the final result. The enhancement result is shown in Figure 6d. Our method also can be applied on other types of sonar image as shown in Table 2. From our experiment $\theta_1 = 50$, $\gamma = 10$ is the shadow threshold value, and $\theta_2 = 180$, $\gamma = 0.5$ is the shadow reflector value.

**Table 2.** Three sonar styles applied on our style transfer process.

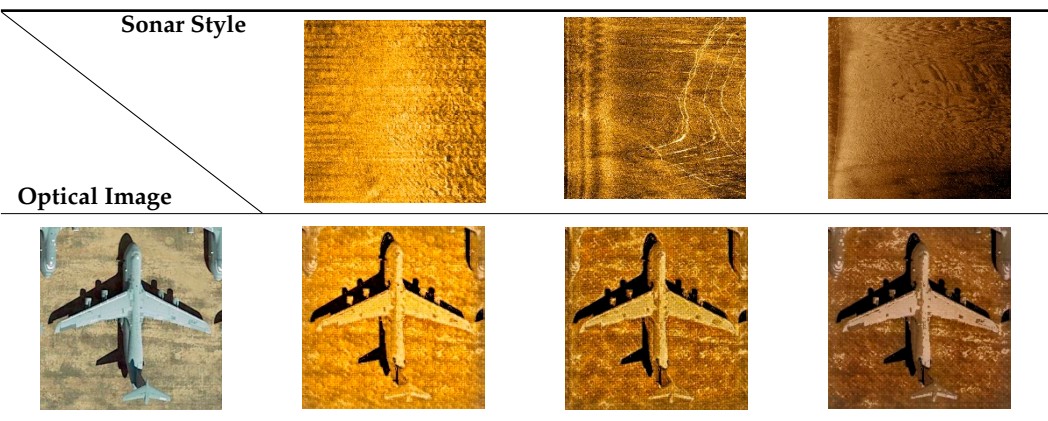

### 3.4. Improved Methods of Designing Random Shape Noise on Target

In the actual sonar image application, it is not difficult to see that many of the targets to be detected are incomplete targets, or some are defective, examples of incomplete target are shown in Figure 12.

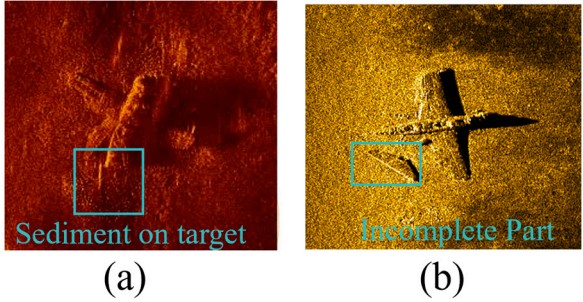

**Figure 12.** Examples of incomplete target. (**a**) Sediment on target. (**b**) Defective target.

On the other hand, benefiting from the rapid development of DCNN, the network can extract object features from data easily. From the zero-shot samples, the network can extract most object features with less real conditions, because the samples which we used for training are too perfect to be close to reality. Our goal is to extract the key features from zero-shot samples and reduce excess features. We propose a method that generates random shape noises on target to simulate the real environment. We defined three types of shapes, points, lines, and rectangles and integrated the classified optical image data to add random noises in our experiment. The noises shape can also be a different type of shape.

We define $P$ as the probability which generates random noise on targets. $P_1$ is the probability of random lines on targets, $N_1$ is the number of lines. $P_2$ is the probability of random point on targets, $N_2$ is the number of points, $P_3$ is the probability of random rectangles on targets, $N_3$ is the number of rectangles, $P_n$ is the probability of random shape on targets, $N_n$ is the number of shapes, $X$ is the total number of optical image samples, $Z$ is the total number of the training data, and $Y$ is the total number of noises. The process of generate noises can be express as follows:

$$Z = X\left(1 + \sum_{i=1}^{n} P_i\right), Y = \sum_{i=1}^{n} P_i * N_i, \ N_i \in (1, 2, 3 \ldots N). \tag{3}$$

Some noises are randomly selected is shown in Tables 3 and 4:

**Table 3.** Noise quantities and type mapping.

| Type \ Quantity | 4 | 6 | 8 |
|---|---|---|---|
| Line |  |  |  |
| Point |  |  |  |
| Rectangle |  |  |  |

**Table 4.** Style transfer result on Table 3.

| Type \ Quantity | 4 | 6 | 8 |
|---|---|---|---|
| Line |  |  |  |
| Point |  |  |  |
| Rectangle |  |  |  |

The area of noise in the target can be express as follows:

$$y = \begin{cases} \left(\frac{\max(w,h)}{16}\right) * l, l = \sqrt[2]{(x_1 - x_2)^2 + (y_1 - y_2)^2}, line\ noise; \\ \left(\frac{w}{5}\right)^2, rectangle\ noise; \\ \pi * r^2, point\ noise. \end{cases} \tag{4}$$

Here, $y$ is the shape noise area in the image, $(w, h)$ is the image's width and height, $l$ is length of noise line, $(x_1, y_1), (x_2, y_2)$ is two connect points for noise line in the image, and $r$ is radius of point noise and value between $\frac{w}{10}$ and $\frac{w}{16}$. All the constant parameters described in the express are the fine-tuned results from our experiments.

On the flip side, overusing the method of shape noise causes low detection performance. Due to the relatively large area of a single noise coverage target, too many noise points are not taken to avoid the problem of excessive coverage, which is shown in Figure 13, the excessive noise almost completely covers the target, resulting in the original image losing target features and decreasing in the detection rate.

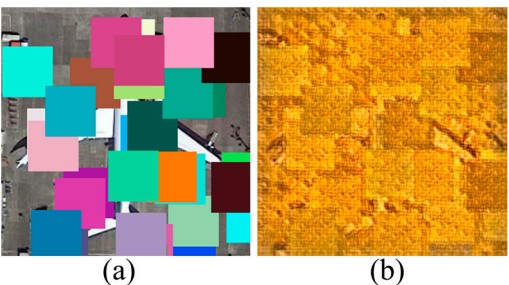

(a)                                    (b)

**Figure 13.** Over noise target. (**a**) Over noise on target. (**b**) Style transfer on target.

## 4. Experiment and Analysis

In this section, we perform a series of experiments to compare the performance of existing methods and our method.

### 4.1. Experiment Data

In order to enrich the diversity of the target forms, our experiment uses part of the VOC2007 dataset [21] and remote-sensing images [22] dataset for training. This paper has conducted a comparative experiment on the same batch of real sonar data. A total of 29 real sonar aircraft wrecks and 43 real shipwreck sonar images were compared and verified under three indicators. Example of sonar images is shown in Figure 14.

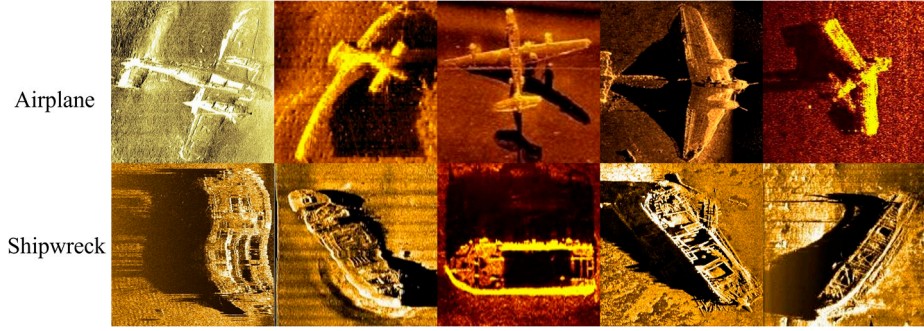

**Figure 14.** Examples of real sonar images.

### 4.2. Experiment Details of Training on Style Transfer Learning and Target Detection Model

We used Python 3.6 and yolov5-large premodel for training. Our training platform employs i7-10700F and GPU NVIDIA GeForce RTX 2060. The content image uses 40,504 images from the COCO 2014 [23] in style transfer training. The training time is 8 h, and the average time spent on one picture by using the style transfer model's transfer is < 0.2 s. The training detection model takes 8 h to detect a $540 \times 480$ pixel sheet containing 3 targets, with an average time < 0.1 s.

Interception of part of the training process is shown in Figure 15, after trained 80 batches, and the results tend to be stable.

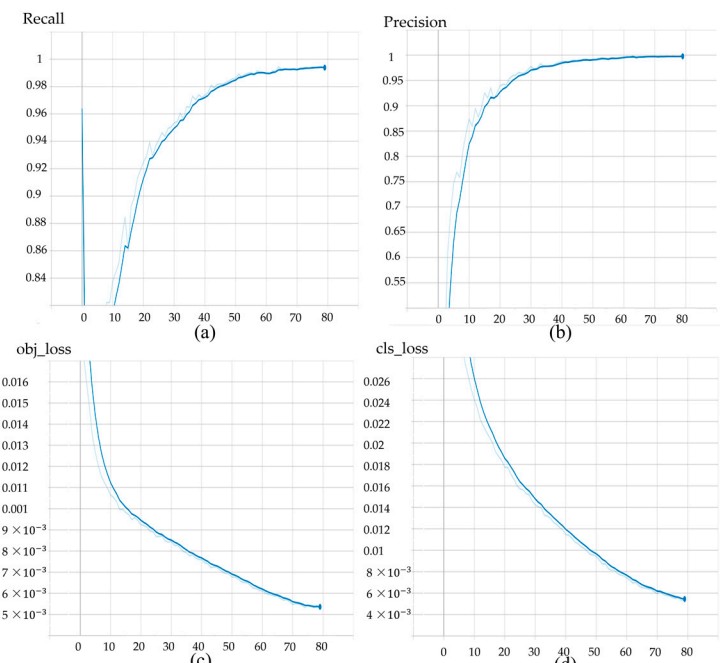

**Figure 15.** Indicator of training. (**a**) Precision trend in training. (**b**) Recall trend in training. (**c**) Object loss is the error caused by confidence. (**d**) Class loss is the error caused by target's type.

Currently, there are two major DCNN-based target detection methods, Faster R-CNN [24] and Yolov5. Faster RCNN has a higher accuracy and lower speed compared with Yolov5. Yolov5 has an easier engineering deployment.

### 4.3. Experiment Results on Comparison with Existing Methods

The proposed method transfers optical images in various datasets into sonar image style images, providing an effective way to enrich target poses. The method of feature enhancement on sample makes the fake-target closer to the real target by using random shape noises on target to simulate the actual environment to enhance the model detection ability.

We use precision and recall as the common criteria for judging target detection models. The definition of precision is that the number of samples be predicted results are correct. Recall refers to the number of positive samples be detected correctly in the predicted results according the published papers and code by existing methods, running in our experiment data. To further evaluate the performance of our method, the comparison with existing methods is shown in Table 5.

From the results, we can see that if the engineer or researcher focus on precision, the proposed method has the better results. From the comparison table, the precision is increased by 0.044 compared with the existing top of precision when using Fastrer R-CNN. Compared with the existing top of precision, the precision is increased by 0.024 when using yolov5.

We combine the style mode and the two major DCNN methods to do experiments on real data is shown in Table 6. The results of this experiment show that the model after adding interference has a higher detection rate and can be used for the detection in sonar images of the real environment.

**Table 5.** Comparison on existing methods performance.

| Model | Precision | Recall | mAP (IOU = 0.5) |
|---|---|---|---|
| StyleBank + fastrcnn [4] | 0.860 | 0.705 | 0.786 |
| Whitening and Coloring Transform [3] | 0.875 | 0.836 | 0.75 |
| Improved style transfer+yolov5 [2] | 0.853 | 0.945 | 0.876 |
| Our method 1: fast style + yolov5 + shape noise | 0.899 | 0.861 | 0.865 |
| Our method 2: fast style + Fastrer R-CNN + shape noise | 0.919 | 0.792 | 0.882 |

**Table 6.** Comparison of model in precision and recall.

| Model | Precision | Recall |
|---|---|---|
| fast style + Yolov5 + shape noise | 0.899 | 0.861 |
| faststyle + Fastrer R-CNN + shape noise | 0.919 | 0.792 |
| fast style + Yolov5 + shape noise and mix real data | 0.957 | 0.944 |
| fast style + Yolov5 + gauss noise | 0.868 | 0.819 |
| fast style + Yolov5 + salt and pepper noise | 0.870 | 0.833 |
| fast style + Yolov5 | 0.873 | 0.764 |
| StyleBank + Yolov5 | 0.755 | 0.563 |
| faststyle + Faster R-CNN | 0.809 | 0.764 |

From the experimental results of Table 6, we can clearly conclude that after adding interference noises to the yolov5 model, the precision is increased by 0.026, and recall is increased by 0.097. After adding interference noises to the Faster R-CNN model, the precision is increased by 0.11, and recall is increased by 0.028. Note that the sonar detection model not using real data on training phase in Tables 6 and 7.

**Table 7.** Detection result with two datasets.

| Model | Dataset | Precision | Recall |
|---|---|---|---|
| style transfer+ yolov5 | remote sensing images + VOC2007 | 0.873 | 0.764 |
| style transfer + yolov5 | remote sensing images | 0.815 | 0.736 |

In additional, we fine tuned the model with mixed real data in optical image samples under the yolov5 model. A total of 72 real sonar images mixed in 1310 optical images. We attained 0.957 precision and 0.944 recall. The result also shown in Table 6. Considering the complex underwater environment, most of the existing methods detect objects and training model without real data. We also cannot use the result to compare existing methods.

*4.4. Further Analysis on Our Methods*

4.4.1. Experiment on Multiples Poses and Shape Noises

The experiment results about combined mutilated poses is shown in Table 7 under the same detection model.

After combining different poses to the yolov5 model, the precision is increased by 0.058, and the recall is increased by 0.028. Experiments show that the model with random noise can detect the incomplete target. The detection result on validation data is shown in Figure 16. By contrast, the model without random noise cannot detect the incomplete target.

The detection results obtained by the number and type of noise are shown in Table 8, which is the recognition rate of the same target. In the aircraft wreck recognition rate under

different noise conditions, the ratio of noise to noiseless is 1:1, and the ratio of mixed line, point, rectangle and noiseless is 1:1:1:1.

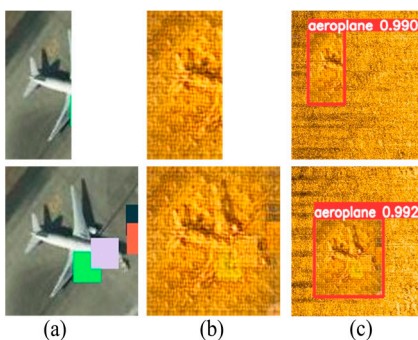

(a)        (b)        (c)

**Figure 16.** Detection result on validation data. (**a**) Simulated incomplete target. (**b**) Style transfer target. (**c**) Detection result in sonar background.

**Table 8.** Detection confidence on 4 types of noises (test in validation data).

| Noise Quantity / Noise Type | Line | Point | Rectangle | Mixed |
|---|---|---|---|---|
| 0 | | | 0.954 | |
| 4 | 0.945 | 0.921 | 0.946 | 0.959 |
| 6 | 0.954 | 0.931 | 0.956 | 0.972 |
| 8 | 0.967 | 0.954 | 0.961 | 0.978 |

It is not difficult to find from Table 8 that the noise detection accuracy of the mixed type is higher than that of a single noise, and the accuracy rate is close when the number of noises is 6 and 8. Some noises are randomly selected is shown in Tables 3 and 4. In this noise design, we define the target image size as long h, width as w pixels, and w > h: (a) rectangular noise length and width as w/5 pixels rectangle. (b) The radius of point noise is a random value between w/16 and w/10 pixels. (c) Line noise is a randomly long line between 5 pixels and w, and a randomly wide line between 2 and 10 pixels. In our experiments, the detection confidence value reached peak when the noise quantities equal to 8. The situation of over noises on target which introduced in end of Section 3.4.

Compared with random shapes noise, we also analyze other types of noise which impact the performance on detection target. This includes Gaussian noise and salt and pepper noise [25]. The result of image noises and style transferred image is shown in Table 9:

**Table 9.** Noises and style transfer image.

| | No-Noise | Gauss Noise | Salt and Pepper |
|---|---|---|---|
| Optical | | | |
| Styled | | | |

The detection compares result with real data is shown in Table 10. Apparently, the shape noise performance is higher than any other types of noise.

**Table 10.** Detection result for 3 types of noise.

| Model | Precision | Recall |
| --- | --- | --- |
| fast style + yolov5 + shape noise | 0.899 | 0.861 |
| fast style + yolov5 + gauss noise | 0.868 | 0.819 |
| fast style + yolov5 + salt and pepper noise | 0.870 | 0.833 |

4.4.2. Experiment on Two Style Models

As described in Section 2 and benefiting from the rapid development of sonar image detection field, style transfer model which has become a common skill of sonar target detection. Fast neural style is the basic style transfer model in this paper due to high performance in our experiments. We compared fast neural style with StyleBank transfer models in our detection work. Figure 17 shows that comparison of the training set and real data under the two style models.

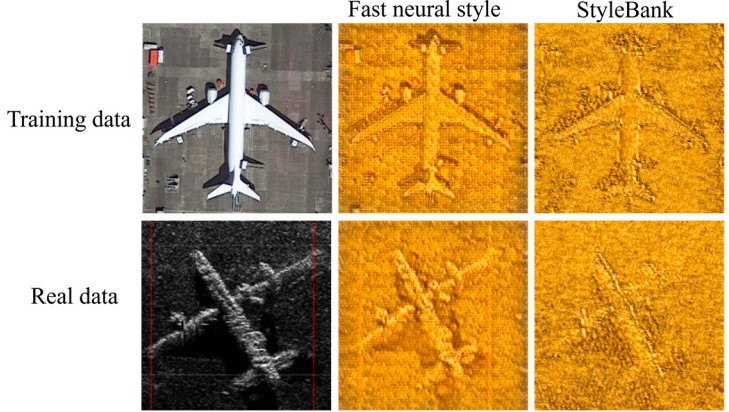

**Figure 17.** Comparison of style models in the training and real data.

From the result of image comparison in Figure 16, we see that fast neural style has a clearer target than StyleBank model. The results of performance comparison between fast neural style and StyleBank is shown in Table 11.

**Table 11.** Detection result for 2 style models.

| Model | Precision | Recall |
| --- | --- | --- |
| fast style + yolov5 | 0.873 | 0.764 |
| StyleBank + yolov5 | 0.755 | 0.563 |

The average confidence value of all detected target under fast neural style model is 0.970. Furthermore, the average confidence value of all detected target under StyleBank model is 0.936. From the results, we know that fast neural style has better performance.

4.4.3. Experiment on Real Data

In order to better approach to the real sonar data in experiment, we perform three types of experiments.

(1)  Detect transferred original image which fusion background style and enhance object feature.

First, perform style transfer of the original image data. Secondly, for simulated target shadow and reflector, we combine the original data with binarization processing and

gamma transformation to simulate real sonar images which are introduced in Section 3.3. Thirdly, we detect the object and the application to the images of real sonar aircraft wreckage process is shown in Figure 18.

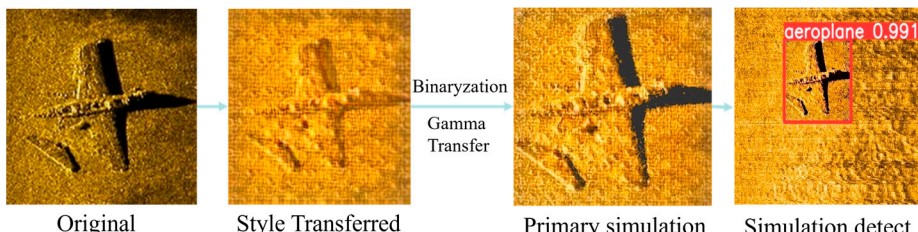

| Original | Style Transferred | | Primary simulation | Simulation detect |

Binaryzation
Gamma Transfer

**Figure 18.** Real data verification process.

From the results, we know that it can be applied to real object detection.

(2)  Detect wreckage data which simulate from real sonar data.

We carried out a wreckage simulation for the real sonar data; the simulation process is shown in Figure 19, which indicates that the detection of the wreckage can be better simulated after adding noise.

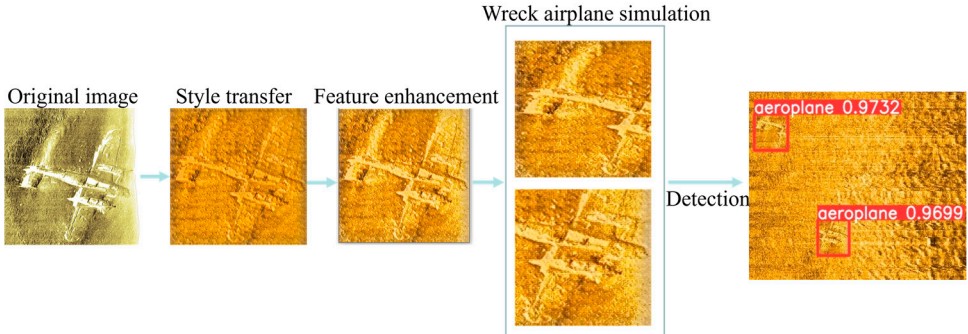

**Figure 19.** Wreckage simulation process and detection result.

At the same time, the real sonar image data is analyzed, and the enhancement methods are added to further restore the real scene for primary simulation.

(3)  Detect real data and difference target size.

Our model can be fit-customized target sizes which can be define in training data. We adjust the target size to (64, 64), (64, 32), (32, 32), (32, 64), and (128,128) in training data. From the results, we know that the detection model has similar performance. An example of difference size on same target is shown in Figure 20.

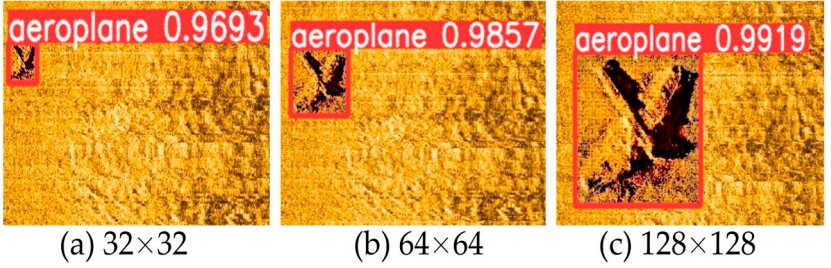

(a) 32×32          (b) 64×64          (c) 128×128

**Figure 20.** Difference size on same target. (**a**) target size is (32,32); (**b**) target size is (64,64); (**c**)target size is (128,128).

It is worth mentioning that the public data of the unprocessed (without style transfer) target and detection result by our proposed method is shown in Figure 21. The detection result by our proposed method has 0.93 confidence value.

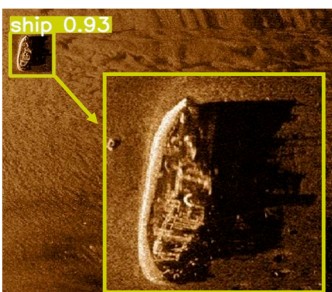

**Figure 21.** Detection on real data (Without Style transfer).

It should be noted that the detection result uses the model without real data on training phase (only optical image in training dataset).

## 5. Conclusions

In this paper, we applied yolov5 and Faster R-CNN, which are used to improve underwater detection performance with lack of training data. We introduce the design consideration of a complex underwater situation and lack of samples, and the limitation of the small area on images with style transfer will cause low performance on object detection. In addition, the designed shapes noises on images solve the problem of mud and sand obstruction in the real environment. At the same time, the combined various optical image datasets enhance multiple poses to solve the issue of the single state of the target. Furthermore, we use binary and gamma method to enhance object features which solve the key features of reflector and shadow are missing when use style transfer directly.

Through the detailed comparison experiment results, we know that the performance of training data with combined mutilated poses is better than without the data. The performance of training data with shape noise is better than Gaussian noise or salt and pepper noise. The performance of faster style transfer better than Stylebank. The performance of Faster R-CNN is better than yolov5. In addition, the selected model can be applied to simulated wreckage data and difference target sizes. We selected the top two with high detection performance in all of our experiments, and we compared it with the existing method to know that the proposed method can achieve better target detection performance than other methods without shape noise fusion and or key feature enhancement in training data.

**Author Contributions:** Conceptualization, J.X. and X.Y.; methodology, J.X.; software, J.X.; validation, J.X., X.Y. and C.L.; formal analysis, J.X.; investigation, J.X. and C.L.; resources, J.X. and X.Y.; data curation, J.X. and X.Y.; writing—original draft preparation, J.X.; writing—review and editing, J.X., X.Y. and C.L.; visualization, J.X.; supervision, X.Y.; project administration, J.X.; funding acquisition, X.Y. All authors have read and agreed to the published version of the manuscript.

**Funding:** This work was supported by the National Natural Science Foundation of China (Grant No. 42276187 and 41876100) and the Fundamental Research Funds for the Central Universities (Grant No. 3072022FSC0401).

**Data Availability Statement:** All the experiments data and code can be found in https://github.com/xijier/SonarDetection (accessed on 22 October 2022).

**Acknowledgments:** We would like to thank the editor and the anonymous reviewers for their valuable comments and suggestions that greatly improve the quality of this paper. Thanks to the researchers who have published datasets of sonar and optical images. https://www.edgetech.com/underwater-technology-gallery/ (accessed on 18 July 2022), https://sandysea.org/projects/visualisierung (accessed on 18 July 2022).

**Conflicts of Interest:** The authors declare no conflict of interest.

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
