# Peer review of "Sonar Image Target Detection Based on Style Transfer Learning and Random Shape of Noise under Zero Shot Target"

_remotesensing, doi:10.3390/rs14246260_

Round 1

Reviewer 1 Report

1.     In this paper, a sonar image target detection method based on style transfer and random shape of noise is proposed. It is for lack of samples and the complicated situation. Also, style transfer is a transfer learning method that pays attention to image texture, but this paper does not mention some basic reasons for using this method in sonar image detection.

2.     270 lines: NVIDIDAshould be ’NVIDIA’

3.     311 lines: This paper refers to two models. Which model has improved the precision and recall after mixing with the real image? The description is not enough and is not shown in the table for comparison.

4.     331 lines: Apart from the situation of no noise, the detection confidence increases with the noise quantity, but this paper does not continue to increase the noise quantity. The reason here may need to be explained.

5.     371 lines: Real-time video detection requires strict time limitation, but this paper does not give a specific time of the detection.

Author Response

Dear Reviwer,

We are appreciated your excelent summarize about our paper. Thank you for your decision and constructive comments on my manuscript.

We have carefully considered the suggestion of Reviewer and make some changes. And we have tried our best to improve and made some changes in the manuscript. The red part that has been revised according to your comments. Revision notes, point-to-point, are given as follows:

  1. In this paper, a sonar image target detection method based on style transfer and random shape of noise is proposed. It is for lack of samples and the complicated situation. Also, style transfer is a transfer learning method that pays attention to image texture, but this paper does not mention some basic reasons for using this method in sonar image detection.

Response 1: Accept. We are appreciated your excelent summarize about our work. And we apologize for not expressing the basic reason about using style transfer clearly. And revised in paper in line 121-124. as below. In difference sonar datasets has difference image textures, and style transfer learning method is pays attention to image texture. To reduce the manual work in difference sonar dataset, the proposed method can be applied difference sonar datasets also.

  1. 270 lines: ‘NVIDIDA’ should be ’NVIDIA’

Response 2: Accept. Thank you very much for finding this error. We are sorry for this mistake. And we revised in paper in line 304.

  1. 311 lines: This paper refers to two models. Which model has improved the precision and recall after mixing with the real image? The description is not enough and is not shown in the table for comparison.

Response 3: Accept. Thanks a lot for the question and suggetions. Revised in paper in line line 351-355. Update experiment results with real image in table 6. fast style + Yolov5 + shape noise and mix real data. Line 345. Also, add the description in paper for the models as below: In additional, we finetuned the model with mixed real data in optical image samples under yolov5 model. 72 real sonar images mixed in 1310 optical images. We got 0.957 precision and 0.944 recall. The result also shown in table 6. Considering complex under-water environment, most of existing methods detect object and training model without real data. We cannot use the result to compare existing methods also.

  1. 331 lines: Apart from the situation of no noise, the detection confidence increases with the noise quantity, but this paper does not continue to increase the noise quantity. The reason here may need to be explained.

Response 4: Accept. Revised in paper as below for the description: In our experiments, the detection confidence value reached peak when the noise quantities equal to 8. The situation of over noises on target which introduced in end of section 3.4. Line 391-393.

  1. 371 lines: Real-time video detection requires strict time limitation, but this paper does not give a specific time of the detection.

Response 5: Accept. We apologize for not considering the strict time limitation about real-time sufficiently. Revised in paper in line 426. Delete the wording of “Real-time video”. Because the real-time cannot be proved in our experiments. Most of our work focus on sonar dataset.

Thanks again for your review, it’s a great help for our work.

BRs,

--Author

Reviewer 2 Report

1.    The authors propose a novel sonar image object detection method based on style learning and random shape of noise. Sonar style target sample images are generated through style transfer, which enhances insufficient sonar objects image. By introducing various noise shapes, the problems of mud and sand obstruction and mutilated target in the real environment are solved, and the single poses of the sonar image target is improved by fusing multiple poses of optical image target.

2.     In the figure 11,. style transfer process should be demonstrated in detail.

3.     The manuscript has 21 figures; the number of the figures should be decreased.

4.     Revise the English thoroughly before submission.

Author Response

Dear Reviwer,

Thank you for your decision and constructive comments on our manuscript. We have carefully considered the suggestion of Reviewer and make some changes.

We have tried our best to improve and made some changes in the manuscript. The red part that has been revised according to your comments. Revision notes, point-to-point, are given as follows:

  1. The authors propose a novel sonar image object detection method based on style learning and random shape of noise. Sonar style target sample images are generated through style transfer, which enhances insufficient sonar objects image. By introducing various noise shapes, the problems of mud and sand obstruction and mutilated target in the real environment are solved, and the single poses of the sonar image target is improved by fusing multiple poses of optical image target.

Response 1: Accept. We are appreciated your excelent summarize about our paper. The summarize included all of our key works.

  1. In the figure 11,. style transfer process should be demonstrated in

Response 2: Accept. Revised in paper to descible style transfer process in line 225-233 for Figure.11(a) and Figure.11(b) in line 239-245.

  1. The manuscript has 21 figures; the number of the figures should be decreased.

Response 3: Your suggestion is valid. It can purify the paper. We have carefully evaluated each figures in these days and hard to cut down these figures. Because each of our figures to make reader understanding the paper clear and more readable. And we will following your suggestions in the further paper work.

  1. Revise the English thoroughly before submission.

Response 4: Accept. We have asked a native English editor to revise the manuscript. In additonal, we will submit the manuscript for English editing at https://www.mdpi.com/authors/english to improve the paper.

Thanks again for your review, it’s a great help for our work.

PS: All the lines which we mentioned in this response review document is by MS word version. If the line is not matched when you review, please let me know or adjust to PDF version review. Thanks for your understanding.

BRs,

--Author

Reviewer 3 Report

1.In the research, three types of regular shape noises, namely points, lines and rectangles, are added to original image, which vary in position and size, but the shape is not completely random. Therefore, the author should consider whether the use of "random shape of noise" is appropriate and accurate. It is also suggested that the shape of the noise should be explained in the abstract.

2. On line 299, "The precision is increased by 0.011 when using yolov5". But I can not find any result 0.011 lower than yolov5 in Table 5. Please confirm the data.

3. The radius of the point noise varies between w/16 and w/10, but the width of the rectangle is fixed at w/5. Why does the width of the rectangle not vary within a certain range?

4. It is suggested to adopt AP to evaluate the precision and accuracy of the target detection models

5. In Table8, it is shown that the more shape noises, the higher the prediction accuracy. At the same time, it is also mentioned that "excessive noise could result in detection performance degradation". Is there an optimal number or area of shape noises?

6. When the abbreviation of proper nouns first appear, you should spell out the acronyms. Such as “CAD” on line 66 and ”WCT” on line 67.

7. Please make sure the accuracy of figure number. Such as “Figure 5(a)” on line 164. It should be figure 11(a).

8. The format of figure captions should be unified and correct. Such as “Figure.8(b)”, “Figure 8.(c) ” and “Figure 11 (b) ”. There are two “(d)” figures on line 186 and line 192. There are some mistakes on line 375 and line 383.

Author Response

Dear Reviwer,

Thank you for your decision and constructive comments on our manuscript. We have carefully considered the suggestion of Reviewer and make some changes.

We have tried our best to improve and made some changes in the manuscript. The red part that has been revised according to your comments. Revision notes, point-to-point, are given as follows:

  1. In the research, three types of regular shape noises, namely points, lines and rectangles, are added to original image, which vary in position and size, but the shape is not completely random. Therefore, the author should consider whether the use of "random shape of noise" is appropriate and accurate. It is also suggested that the shape of the noise should be explained in the abstract.

Response 1: Accept. It’s very good suggestions. The shapes is selected by us in our experiments and noises is generated by randomly. We should explain "random shape of noise" in our paper also. As below is modifed and explaination and revised in abstract. Revised "random shape of noise" to “random of noise with various shapes.” in Line 14. And the shape of the noise revised in abstract. "By introducing various noise shapes which included points, lines and rectangles." Line 16.

  1. On line 299, "The precision is increased by 0.011 when using yolov5". But I can not find any result 0.011 lower than yolov5 in Table 5. Please confirm the data.

Response 2: Accept. Thanks a lot for the error and we apologize for the typo error in caluation. Revised in the paper. In additonal, we also add the sentence for the caluation in line 341 “Compared with the existing top of precision.” make it more clearly.

  1. The radius of the point noise varies between w/16 and w/10, but the width of the rectangle is fixed at w/5. Why does the width of the rectangle not vary within a certain range?

Response 3: Accept. We apologize for not expressing ourselves clearly. Revised in pager in line 282-283. All the parameters which described in the express is the finetune results from our experiments from our experiments. The area of noises which covered in the target has infulences detection performance. We are not doing detail anlysis about the correlation analysis between covered area and detection performance. Due to we got the experienced constant value. We will doing more research in the following work and paper.

  1. It is suggested to adopt AP to evaluate the precision and accuracy of the target detection models.

Response 4: Accept. That is a very good and reasonable suggestion.Revised in paper line 337 in table 5.

  1. In Table8, it is shown that the more shape noises, the higher the prediction accuracy. At the same time, it is also mentioned that "excessive noise could result in detection performance degradation". Is there an optimal number or area of shape noises?

Response 5: Accept. Revised description in paper line 393-395. In our experiments, the detection confidence value reached peak when the noise quantities equal to 8. The situation of over noises on target which introduced in end of section 3.4.

  1. When the abbreviation of proper nouns first appear, you should spell out the acronyms. Such as “CAD” on line 66 and ”WCT” on line 67.

Response 6: Accept. We apologize for not notice the details. Revised in paper line 66 and line 67.

  1. Please make sure the accuracy of figure number. Such as “Figure 5(a)” on line 164. It should be figure 11(a).

Response 7: Accept. Thank you very much for finding this error. We are sorry for this problem and have corrected it according to your suggestion. Revised in paper line 225. Also, we realigned the descripiton which should near the figure 11.

  1. The format of figure captions should be unified and correct. Such as “Figure.8(b)”, “Figure 8.(c) ” and “Figure 11 (b) ”. There are two “(d)” figures on line 186 and line 192. There are some mistakes on line 375 and line 383.

Response 8: Accept. Revised in paper. Line 183, 214. In addiation, we have go though the paper which related format of figure.

Thanks again for your review, it’s a great help for our work.

PS: All the lines which we mentioned in this response review document is by MS word version. If the line is not matched when you review, please let me know or adjust to PDF version review. Thanks for your understanding.

BRs,

--Author

Reviewer 4 Report

Comments on the manuscript “Sonar image target detection base don style transfer learning and random shape of noise under zero shot target “by Jier Xi et al.

This is a very interesting manuscript.  The presented work applies a method based on learning and random shape of noise to improve sonar object detection and recognition algorithms currently used. To do it, authors solve the problem of the lack of real sonar data generating sonar images through style transfer and by introducing different noise shapes in order to reproduce obstructions and mutilated images due to  mud and sand effect. Moreover, multiple poses of optical images is used to feed the sonar image database. The results presented by authors show good target detection with a high confidence values.

In general, this is a very well written and well-structured manuscript. The manuscript is suitable for publication in the journal. In general, this is a very well written and well-structured manuscript. The manuscript is suitable for publication in the journal. This reviewer has minor comments on the manuscript:

Lines 162-163: General process is insufficiently described in my opinion. A new wording is proposed for this paragraph.

Figure 15: Values in x and y axes cannot be read. This figure should be remade in order to allow the reader understand it.

The Conclusions section must be improve. A more detailed comparison of the results of all methods is lacking in this part.

Would be very welcome that the minor issues exposed, will be better explained and clarified in detail to the better understanding of results depicted, and applied methodology.

Author Response

Dear Reviwer,

We are appreciated your excelent summarize about our paper. Thank you for your decision and constructive comments on our manuscript.

We have carefully considered the suggestion of reviewer and make some changes. And we have tried our best to improve and made some changes in the manuscript. The red part that has been revised according to your comments. Revision notes, point-to-point, are given as follows:

  1. This Lines 162-163: General process is insufficiently described in my opinion. A new wording is proposed for this paragraph.

Response 1: Accept. That is a very good and reasonable suggestion. A new wording is revised in this paragraph to make the gerneral process more straightforward. Line 169-171. In addation, to describe the general process sufficiently, we have adjust part of this paragraph and wording near to Figure.11(a) to make the gerneral process more understandable in line 226-234.

  1. Figure 15: Values in x and y axes cannot be read. This figure should be remade in order to allow the reader understand it.

Response 2: Accept. Thank you very much for finding this error. Remade Figure 15 and Revised in paper in line 343.

  1. The Conclusions section must be improve. A more detailed comparison of the results of all methods is lacking in this part.

Response 3: Accept. We apologize for not expressing the detailed comparison clearly in conclusion section. Revised in paper line 449-507 as below. Through the detail comparison experiment results knows that the performance of training data with combined mutilate poses better than without the data. The performance of training data with shape noise better than gauss, salt and pepper noise. The perfor-mance of faster style transfer better than Stylebank. The performance of Faster R-CNN bet-ter than yolov5. In addition, the selected model can be applied simulated wreckage data and difference target sizes. We selected the top two with high detection performance in all of our experiments, and compare with existing method knows that the proposed method can achieve better target detection performance than other methods which without shape noises fusion and no key feature enhancement in training data.

Thanks again for your review, it’s a great help for our work.

PS: All the lines which we mentioned in this response review document is by MS word version. If the line is not matched when you review, please let me know or adjust to PDF version review. Thanks for your understanding.

BRs,

--Author

Round 2

Reviewer 1 Report

All my comments on this paper have been answered correctly. I do not have further comments. I think this paper can be accepted in its present form.